# Dynamic Network Model from Partial Observations

**Elahe Ghalebi**
TU Wien
eghalebi@cps.tuwien.ac.at

**Baharan Mirzasoleiman**
Stanford University
baharanm@cs.stanford.edu

**Radu Grosu**
TU Wien
radu.grosu@tuwien.ac.at

**Jure Leskovec**
Stanford University
jure@cs.stanford.edu

## Abstract

Can evolving networks be inferred and modeled without directly observing their nodes and edges? In many applications, the edges of a dynamic network might not be observed, but one can observe the dynamics of stochastic cascading processes (e.g., information diffusion, virus propagation) occurring over the unobserved network. While there have been efforts to infer networks based on such data, providing a *generative probabilistic model* that is able to identify the underlying time-varying network remains an open question. Here we consider the problem of inferring generative dynamic network models based on network cascade diffusion data. We propose a novel framework for providing a non-parametric dynamic network model—based on a mixture of coupled hierarchical Dirichlet processes—based on data capturing cascade node infection times. Our approach allows us to infer the evolving community structure in networks and to obtain an explicit predictive distribution over the edges of the underlying network—including those that were not involved in transmission of any cascade, or are likely to appear in the future. We show the effectiveness of our approach using extensive experiments on synthetic as well as real-world networks.

## 1 Introduction

Networks of interconnected entities are widely used to model pairwise relations between objects in many important problems in sociology, finance, computer science, and operations research [1, 2, 3]. Often times, these networks are dynamic, with nodes or edges appearing or disappearing over time, and the underlying network structure evolving over time. As a result, there is a growing interest in developing dynamic network models allowing for the study of evolving networks.

Non-parametric models are specially useful when there is no prior knowledge or assumption about the shape or size of the network as they can automatically address the model selection problem. Non-parametric Bayesian approaches mostly rely on the assumption of vertex exchangeability, in which the distribution of a graph is invariant to the order of its vertices [4, 5, 6]. Vertex-exchangeable models such as the Stochastic Block model and its variants, explain the data by means of an underlying latent clustering structure. However, such models yield dense graphs [7] and are less appropriate for predicting unseen interactions. Recently, an alternative notion of edge-exchangeability was introduced for graphs, in which the distribution of a graph is invariant to the order of its edges [8, 9]. Edge-exchangeable models can exhibit sparsity, and small-world behavior of real-world networks. Such models allow both the latent dimensionality of the model and the number of nodes to grow over time, and are suitable for predicting future interactions.

Existing models, however, aim to model a fully observed network [4, 5, 8, 9] but in many real-world problems, the underlying network structure is not known. What is often known are partial observations of a stochastic cascading process that is spreading over the network. A cascade is created by a contagion (e.g., a social media post, a virus) that starts at some node of the network and then spreads like an epidemic from node to node over the edges of the underlying (unobserved) network. The observations are often in the form of the times when different nodes get infected by different contagions. A fundamental problem, therefore, is to infer the underlying network structure from these partial observations. In recent years, there has been a body of research on inferring diffusion networks from node infection times. However, these efforts mostly rely on a fixed cascade transmission model—describing how nodes spread contagions—to infer the set of most likely edges [2, 10, 11, 12]. More recently, there have been attempts to predict the transmission probabilities from infection times, either by learning node representations [13], or by learning diffusion representations using the underlying network structure [13, 14, 15]. However, it remains an open problem to provide a generative probabilistic model for the underlying network from partial observations.

Here we propose a novel online dynamic network inference framework, DYFERENCE, for providing non-parametric edge-exchangeable network models from partial observations. We build upon the non-parametric network model of [8], namely MDND, that assumes that the network clusters into groups and then places a mixture of Dirichlet processes over the outgoing and incoming edges in each cluster while coupling the network using a shared discrete base measure. However, our framework is easily extended to arbitrary generative models replacing the MDND with other choices of latent representations, such as network models presented in [9, 16, 17, 18]. Given a set of cascades spreading over the network, we process observations in time intervals. For each time interval we first find a probability distribution over the cascade diffusion trees that may have been involved in each cascade. We then calculate the marginal probabilities for all the edges involved in the diffusion trees. Finally, we sample a set of edges from this distribution and provide the sampled edges to a Gibbs sampler to update the model variables. In the next iteration, we use the updated edge probabilities provided by the model to update the probability distributions over edges supported by each cascade. We continue the above iterative process until the model does not change considerably. Extensive experiments on synthetic and real-world networks show that DYFERENCE is able to track changes in the structure of dynamic networks and provides accurate online estimates of the time-varying edge probabilities for different network topologies. We also apply DYFERENCE for diffusion prediction and predicting the most influential nodes in Twitter and MemeTracker datasets, as well as bankruptcy prediction in a financial transaction network.

## 2   Related Work

There is a body of work on inferring diffusion network from partial observations. NETINF [19] and MULTITREE [20] formulate the problem as submodular optimization. NETRATE [21] and CONNIE [2] further infer the transmission rates using convex optimization. INFOPATH [22] considers inferring varying transmission rates in an online manner using stochastic convex optimization. The above methods assume that diffusion rates are derived from a predefined parametric probability distribution. In contrast, we don't make any assumption on the transmission model. EMBEDDEDIC [13] embeds nodes in a latent space based on Independent Cascade model, and infer diffusion probabilities based on the relative node positions in the latent space. DEEPCAS [15] and TOPOLSTM [14] use the network structure to learn diffusion representations and predict diffusion probabilities. Our work is different in nature to the existing methods in that we aim at providing a generative probabilistic model for the underlying dynamic network from diffusion data.

There has also been a growing interest in developing probabilistic network models that can capture network evolutions from full observations. Bayesian generative models such as the stochastic block model [4], the mixed membership stochastic block model [23], the infinite relational model [24, 25], the latent space model [26], the latent feature relational model [27], the infinite latent attribute model [17, 28], and the random function model [7] are among the vertex-exchangeable examples. A limitation of the vertex-exchangeable models is that they generate dense or empty networks with probability one [29, 30]. This is in contrast with the sparse nature of many real-world networks. Recently, edge-exchangeable models have been proposed and shown to exhibit sparsity [8, 9, 16]. However, these models assume that networks are fully observed. In contrast, our work here considers the network is unobserved but what we observe are node infection times of a stochastic cascading process spreading over the network.

# 3  Preliminaries

In this section, we first formally define the problem of dynamic network inference from partial observations. We then review the non-parametric edge-exchangeable network model of [8] that we will build upon in the rest of this paper. Finally, we give a brief overview of Bayesian inference for inferring latent model variables.

## 3.1  Dynamic Network Inference Problem

Consider a hidden directed dynamic network where nodes and edges may appear or disappear over time. At each time step $t$, the network $G^t = (V^t, E^t)$ consists of a set of vertices $V^t$, and a set of edges $E^t$. A set $C$ of cascades spread over edges of the network from infected to non-infected nodes. For each cascade $c \in C$, we observe a sequence $t^c := (t_1^c, \cdots, t_{|V|}^c)$, recording the times when each node got infected by the cascade $c$. If node $u$ is not infected by the cascade $c$, we set $t_u^c = \infty$. For each cascade, we only observe the time $t_u^c$ when node $u$ got infected, but not what node and which edge infected node $u$. Our goal is to infer a model $\mathcal{M}$ to capture the latent structure of the network $G^t$ over which cascades propagated, using these partial observations. Such a model, in particular, can provide us with the probabilities of all the $|V^t|^2$ potential edges between nodes $u, v \in V^t$.

## 3.2  Non-parametric Edge-exchangeable Network Model

We adopt the Bayesian non-parametric model of [8] that combines structure elucidation with predictive performance. Here, the network is modeled as an exchangeable sequence of directed edges, and can grow over time. More specifically, each community in the network is modeled by a mixture of Dirichlet network distributions (MDND).

The model $\mathcal{M}$ can be described as:

$$
\begin{aligned}
D &:= (d_k, k \in \mathbb{N}) && \sim \mathrm{GEM}(\alpha) \\
H &:= \sum_{i=1}^{\infty} h_i \delta_{\theta_i} && \sim \mathrm{DP}(\gamma, \Theta) \\
A_k &:= \sum_{i=1}^{\infty} a_{k,i} \delta_{\theta_i} && \sim \mathrm{DP}(\tau, H), \quad k = 1, 2, \cdots \\
B_k &:= \sum_{i=1}^{\infty} b_{k,i} \delta_{\theta_i} && \sim \mathrm{DP}(\tau, H)
\end{aligned}
\qquad
\begin{aligned}
&\varsigma_{uv} \sim D, \quad u, v \in V \\
&u \sim A_{\varsigma_{uv}} \\
&v \sim B_{\varsigma_{uv}} \\
&z_{uv} = \sum_{i,j \in V} \mathbb{I}(i = u, j = v),
\end{aligned}
\tag{1}
$$

The edges of the network are modeled by a Dirichlet distribution $H$. Here, $\Theta$ is the measurable space, $\delta_{\theta_i}$ denotes an indicator function centered on $\theta_i$, and $h_i$ is the corresponding probability of an edge to exist at $\theta_i$, with $\sum_{i=1}^{\infty} h_i = 1$. The concentration parameter $\gamma$ controls the number of edges in the network, with larger $\gamma$ results in more edges. The size and number of communities are modeled by a stick-breaking distribution $\mathrm{GEM}(\alpha)$ with concentration parameter $\alpha$. For every community $k$, two Dirichlet distribution $A_k$, and $B_k$ models the outlinks and inlinks in community $k$. To ensure that outlinks and inlinks are defined on the same set of locations $\theta$, distributions $A_k$, and $B_k$ are coupled using the shared, discrete base measure $H$.

To generate an edge $e_{uv}$, we first select a cluster $\varsigma_{uv}$ according to $D$. We then select a pair of nodes $u$ and $v$ according to the cluster-specific distributions $A_{\varsigma_{uv}}, B_{\varsigma_{uv}}$. The concentration parameter $\tau$ controls the overlap between clusters, with smaller $\tau$ results in smaller overlaps. Finally, $z_{ij}$ is the integer-valued weight of edge $e_{ij}$.

## 3.3  Bayesian Inference

Having specified the model $\mathcal{M}$ in terms of the joint distribution in Eq. 1, we can infer the latent model variables for a fully observed network using Bayesian inference. In the full observation setting where we can observe all the edges in the network, the posterior distribution of the latent variables conditioned on a set of observed edges $X$ can be updated using the Bayes rule:

$$
p(\Phi|X, \mathcal{M}) = \frac{p(X|\Phi, \mathcal{M}) p(\Phi|\mathcal{M})}{\int p(X, \Phi|\mathcal{M}) d\Phi}
\tag{2}
$$

Here, $\Phi$ is the infinite dimensional parameter vector of the model $\mathcal{M}$ specified in Eq. 1. The denominator in the above equation is difficult to handle as it involves summation over all possible parameter values. Consequently, we need to resort to approximate inference. In Section 4, we show how we extract our set of observations from diffusion data and construct a collapsed Gibbs sampler to update the the posterior distributions of latent variables.

## 4   DYFERENCE: Dynamic Network Inference from Partial Observations

In this section, we describe our algorithm, DYFERENCE, for inferring the latent structure of the underlying dynamic network from diffusion data. DYFERENCE works based on the following iterative idea: in each iteration we (1) find a probability distribution over all the edges that could be involved in each cascade; (2) Then we sample a set of edges from the probability distribution associated with each cascade, and provide the sampled edges as observations to a Gibbs sampler to update the posterior distribution of the latent variables of our non-parametric network model. We start by explaining our method on a static directed network, over which we observe a set $C$ of cascades $\{t^{c_1}, \cdots, t^{c_{|C|}}\}$. In Section 4.3, we shall then show how we can generalize our method to dynamic networks.

### 4.1   Extracting Observations from Diffusion Data

The set of edges that could have been involved in transmission of a cascade $c$ is the set of all edges $e_{uv}$ for which $u$ is infected before $v$, i.e., $E_c = \{e_{uv} | t^c_u < t^c_v < \infty\}$. Similarly, $V_c = \{u | t^c_u < \infty\}$ is the set of all infected nodes in cascade $c$. To find the probability distribution over all the edges in $E_c$, we first assume that every infected node in cascade $c$ gets infected through one of its neighbors, and therefore each cascade propagates as a directed tree. For a cascade $c$, each possible way in which the cascade could spread over the underlying network $G$ creates a tree. To calculate the probability of a cascade to spread as a tree $T$, we use the following Gibbs measure [31],

$$p(T|\boldsymbol{d}/\lambda) = \frac{1}{Z(\boldsymbol{d}/\lambda)} e^{-\sum_{e_{uv} \in T} d_{uv}/\lambda}, \qquad (3)$$

where $\lambda$ is the temperature parameter. The normalizing constant $Z(\boldsymbol{d}/\lambda) = \sum_{e_{uv} \in E_c} e^{-d_{uv}/\lambda}$ is the partition function that ensures that the distribution is normalized, and $d_{uv}$ is the weight of edge $e_{uv}$. The most probable tree for cascade $c$ is a MAP configuration for the above distribution, and the distribution will concentrate on the MAP configurations as $\lambda \to 0$.

To calculate the probability distribution over the edges in $E_c$, we use the result of [32] who showed that the probability distribution over subsets of edges associated with all the spanning trees in a graph is a Determinantal Point Processes (DPP), where the probability of every subset $R \subseteq T$ can be calculated as:

$$\mathbb{E}_{P(T|\boldsymbol{d}/\lambda)}[\llbracket R \subseteq T \rrbracket] = \det K_R. \qquad (4)$$

Here, $K_R$ is the $|R| \times |R|$ restriction of the DPP kernel $K$ to the entries indexed by elements of $R$. For constructing the kernel matrix $K$, we take the incidence matrix $A \in \{-1, 0, +1\}^{|V_c-1| \times |E_c|}$, in which $A_{ij} \in \{1, 0, -1\}$ indicates that edge $j$ is an outlink/inlink of node $i$, and we removed an arbitrary vertex from the graph. Then, construct its Laplacian $L = A \operatorname{diag}(e^{-\boldsymbol{d}/\lambda}) A^T$ and compute $H = L^{-1/2} A \operatorname{diag}(e^{-\boldsymbol{d}/2\lambda})$ and $K = H^T H$.

Finally, the marginal probabilities of an edge $e_{uv}$ in $E_c$ can be calculated as:

$$p(e_{uv}|\boldsymbol{d}/\lambda) = e^{-d_{uv}/\lambda}(a_u - a_v)^T L^{-1}(a_u - a_v), \qquad (5)$$

where $a_i$ is the vector with coordinates equal to zero, except the $i$-th coordinate which is one. All marginal probabilities can be calculated in time $\tilde{O}(r|V_c|^2/\epsilon^2)$, where $\epsilon$ is the desired relative precision and and $r = \frac{1}{\lambda}(\max_e d(e) - \min_e d(e))$ [33].

To construct our multiset of observations $X$—in which each edge can appear multiple times—, for each $c$ we sample a set $S_c$ of $q$ edges from the probability distributions of edges in $E_c$. I.e,.

$$X = \{S_{c_1}, \cdots, S_{c_{|C|}}\} \qquad (6)$$

Note that an edge could be sampled multiple times from the probability distributions corresponding to multiple cascades. The number of times each edge $e_{uv}$ is sampled is the integer valued weight

---

**Algorithm 1** EXTRACT_OBSERVATIONS

---

**Input:** Set of cascades $\{t^{c_1}, \cdots, t^{c_{|C|}}\}$, sample size $q$.
**Output:** Extracted multiset of edges $X$ from cascades.
1: $X \leftarrow \{\}$
2: **for** $c \in C$ **do**
3:      Calculate $p(e_{uv}|\boldsymbol{d}/\lambda)$ for all $e_{uv} \in E_c$ using Eq. 5
4:      $S_c \leftarrow$ Sample $q$ edges from the above probability distribution.
5:      $X \leftarrow \{X, S_c\}$
6: **end for**

---

$z_{ij}$ in Eq. 1. Initially, without any prior knowledge about the structure of the underlying network, we initialize $d_{uv} \propto \sum_{c \in C} t_v^c - t_u^c$ for all $e_{uv} \in E_c$, and $d_{uv} = 0$ otherwise. However, in the subsequent iterations when we get the updated posterior probabilities from our model, we use $d_{uv} = p(e_{uv}|\Phi, \mathcal{M})$.

The pseudocode for extracting observations from diffusion data is shown in Algorithm 1.

## 4.2 Updating Latent Model Variables

To update the posterior distribution of the latent model variables conditioned on the extracted observations, we construct a collapsed Gibbs sampler by sweeping through each variable to sample from its conditional distribution with the remaining variables fixed to their current values.

**Sampling cluster assignments $\varsigma$.** Following [8], we model the posterior probability for an edge $e_{uv}$ to belong to cluster $k$ as a function of the importance of the cluster in the network, the importance of $u$ as a source and $v$ as a destination in cluster $k$, as well as the importance of $u, v$ in the network. To this end, we measure the importance of a cluster by the total number of its edges, i.e., $\eta_k = \sum_{u,v \in V}[\mathbb{I}_{\varsigma_{uv}} = k]$. Similarly, the importance of $u$ as a source, and the importance of $v$ as a destication in cluster $k$ is measured by the number of outlinks of $u$ associated with cluster $k$, i.e. $l_{u.}^{(k)}$, as well as inlinks of $v$ associated with cluster $k$, i.e. $l_{.v}^{(k)}$. Finally, the importance $\beta$ of node $i$ in the network is determined by the probability mass of its outlinks $h_{i.}$ and inlinks $h_{.i}$, i.e. $\beta_i = \sum h_{i.} + \sum h_{.i}$. The distribution over the cluster assignment $\varsigma_{uv}$ of an edge $e_{uv}$, given the end nodes $u, v$, the cluster assignments for all other edges, and $\beta$ is given by:

$$p(\varsigma_{uv} = k|u, v, \varsigma_{1:M}^{\neg e_{uv}}, \beta_{1:N}) \propto \begin{cases} \eta_k^{\neg e_{uv}}(l_{u.}^{(k)\neg e_{uv}} + \tau \beta_u)(l_{.v}^{(k)\neg e_{uv}} + \tau \beta_v) & \text{if } \eta_k^{\neg e_{uv}} > 0 \\ \alpha \tau^2 \beta_u \beta_v & \text{if } \eta_k^{\neg e_{uv}} = 0 \end{cases} \quad (7)$$

where $\neg e_{uv}$ is used to exclude the variables associated with the current edge being observed. As discussed in Section 3.2, $\alpha$, $\tau$, and $\gamma$ controls the number of clusters, cluster overlaps, and the number of nodes in the network. Moreover, $N, M$ are the number of nodes and edges in the network.

**Sampling edge probabilities $e$.** Due to the edge-exchangeability, we can treat $e_{uv}$ as the last variable being sampled. The conditional posterior for $e_{uv}$ given the rest of the variables can be calculated as:

$$p(e_{uv} = e_{ij}|\varsigma_{1:M}, e_{1:M}^{\neg e_{uv}}, \beta_{1:N}) =$$

$$\begin{cases} \sum_{k=1}^{K+} \frac{\eta_k}{M+\alpha} \frac{l_{u.}^{(k)\neg e_{uv}} + \tau \beta_u}{\eta_k + \tau} \frac{l_{.v}^{(k)\neg e_{uv}} + \tau \beta_v}{\eta_k + \tau} + \frac{\alpha}{M+\alpha} \beta_u \beta_v & \text{if } i, j \in V \\ \sum_{k=1}^{K+} \frac{\eta_k}{M+\alpha} \frac{l_{u.}^{(k)\neg e_{uv}} + \tau \beta_u}{\eta_k + \tau} \beta_n + \frac{\alpha}{M+\alpha} \beta_u \beta_n & \text{if } i \in V, j \notin V \\ \sum_{k=1}^{K+} \frac{\eta_k}{M+\alpha} \frac{l_{.v}^{(k)\neg e_{uv}} + \tau \beta_u}{\eta_k + \tau} \beta_n + \frac{\alpha}{M+\alpha} \beta_n \beta_v & \text{if } i \notin V, j \in V \\ \beta_n^2 & \text{if } i, j \notin V \end{cases} \quad (8)$$

where $\beta_n = \sum_{i=N+1}^{\infty} h_i$ is the probability mass for all the edges that may appear in the network in the future, and $K$ is number of clusters. We observe that an edge may appear between existing nodes in the network, or because one or two nodes has appeared in the network. Note that the predictive distribution for a new link to appear in the network can be calculated similarly using Eq. 8.

---

**Algorithm 2** UPDATE_NETWORK_MODEL

---

**Input:** Model $\mathcal{M}(c_{1:M}, p_{1:M}, \beta_{1:N})$, set of cascades $\{t^{c_1}, \cdots, t^{c_{|C|}}\}$.
**Output:** Updated model $\mathcal{M}^*(c_{1:M}, p_{1:M}, \beta_{1:N})$
  1: **for** $i = 1, 2, \cdots$ until convergence **do**
  2:      $X \leftarrow$ Extract_Observations($\{t^{c_1}, \cdots, t^{c_{|C|}}\}$)
  3:      **for** $j = 1, 2, \cdots$ until convergence **do**
  4:          Select $e_{uv}$ randomly from $X$
  5:          Sample $\varsigma$ from the conditional distribution $p(c_{uv} = k | u, v, \varsigma_{1:M}^{\neg e_{uv}}, \beta_{1:N})$         $\triangleright$ Eq. 7
  6:          Sample $e$ from the conditional distribution $p(e_{uv} = e_{ij} | \varsigma_{1:M}, e_{1:M}^{\neg e_{uv}}, \beta_{1:N})$      $\triangleright$ Eq. 8
  7:          Sample $\rho$ from the conditional distribution $p(\rho_{u.}^{(k)} = \rho | \varsigma_{1:M}, \rho_{u.}^{(k)\neg e_{uv}}, \beta_{1:N})$    $\triangleright$ Eq. 9
  8:          Sample $(\beta_1, \cdots, \beta_{|V|}, \beta_u) \sim \text{Dir}(\rho_1^{(.)}, \cdots, \rho_{|V|}^{(.)}, \gamma)$           $\triangleright$ Eq. 10
  9:      **end for**
10: **end for**

---

**Sampling outlink and inlink probabilities $\rho$.** The probability mass on the outlinks and inlinks of node $i$ associated with cluster $k$ are modeled by variables $\rho_{i.}^{(k)}$ and $\rho_{.i}^{(k)}$. The posterior distribution of $\rho_{u.}^{(k)}$ (similarly $\rho_{.v}^{(k)}$), can be calculated using:

$$p(\rho_{u.}^{(k)} = \rho | c_{1:M}, \rho_{u.}^{(k)\neg e_{uv}}, \beta_{1:N}) = \frac{\Gamma(\tau\beta_u)}{\Gamma(\tau\beta_u + l_{u.}^{(k)})} s(l_{u.}^{(k)}, \rho)(\tau\beta_u)^{\rho}, \tag{9}$$

where $s(l_{u.}^{(k)}, \rho)$ are unsigned Stirling numbers of the first kind. I.e., $s(0,0) = s(1,1) = 1, s(n,0) = 0$ for $n > 0$ and $s(n, m) = 0$ for $m > n$. Other entries can be computed as $s(n+1, m) = s(n, m-1) + ns(n, m)$. However, for large $l_{u.}^{(k)}$, it is often more efficient to sample $\rho_{k,i}$ by simulating the table assignments of the Chinese restaurant according to Eq. 8 [34].

**Sampling node probabilities $\beta$.** Finally, the probability of each node is the sum of the probability masses on its edges and is modeled by a Dirichlet distribution, i.e.,

$$(\beta_1, \cdots, \beta_N, \beta_n) \sim \text{Dir}(\rho_1^{(.)}, \cdots, \rho_N^{(.)}, \gamma), \tag{10}$$

where $\rho_i^{(.)} = \sum_k \rho_{i.}^{(k)} + \rho_{.i}^{(k)}$.

The pseudocode for inferring the latent network variables from diffusion data is given in Algorithm 2.

### 4.3 Online Dynamic Network Inference

In order to capture the dynamics of the underlying network and keep the model updated over time, we consider time intervals of length $w$. For the $i$-th interval, we only consider the infection times $t^c \in [(i-1)w, iw)$ for all $c \in C$, and update the model conditioned on the observations in the current time interval. Updating the model over intervals resembles the continuous time updates with larger steps. Indeed, we can update the model in a continuous manner upon observing every infected node ($w = dt$). However, the observations provided to the Gibbs sampler from a single infection time are limited to the direct neighborhood of the infected node. This increases the overall mixing time as well as the probability of getting stuck in a local optima. Updating the model using very small intervals has the same effect by providing the Gibbs sampler with limited information about directed neighborhoods of few infected nodes.

Note that we do not infer a new model for the network based on infection times in each time interval. Instead, we use new observations to *update* the latent variables from the previous time interval. Updating the model with observations in the current interval results in a higher probability for the observed edges, and a lower probability for the edges that have not been observed recently. Therefore, we do not need to consider an aging factor to take into account the older cascades. For a large $w$, the model may change considerably from one interval to the next. Hence, updating the model from previous interval may harm the solution quality. However, if $w$ is not very large, initializing the parameters from the previous interval significantly improves the running time, while the quality of the solutions are preserved.

---
**Algorithm 3** DYNAMIC_NETWORK_INFERENCE (DYFERENCE)
---
**Input:** Set of infection times $\{t^{c_1}, \cdots, t^{c_{|C|}}\}$, interval length $w$.
**Output:** Updated network model $\mathcal{M}^t$ at times $t = iw$.
 1: $t = w$, initialize $\mathcal{M}^0$ randomly.
 2: **while** $t <$ last infection time **do**
 3:     **for all** $c \in C$ **do**
 4:         $t^{cw} \leftarrow t^c_u \in [t - w, t)$
 5:         $Y^t \leftarrow \{Y^t, t^{cw}\}$
 6:     **end for**
 7:     $\mathcal{M}^t \leftarrow$ Update_Network_Model$(\mathcal{M}^{t-w}, Y^t)$
 8:     $t = t + w$.
 9: **end while**
---

Finally, a very large sample size $q$ provides the Gibbs sampler with uninformative observations, including edges with a low probability, and result in an increased mixing time. Since the model from previous interval has the information about all the infections happened so far, if $w$ and $q$ are not too large, we expect the parameters to change smoothly over the intervals. We observed that $q = \Theta(|E_c|)$ works well in practice.

The pseudocode of our dynamic inference method is given in Algorithm 3.

## 5 Experiments

In this section, we address the following questions: (1) What is the predictive performance of DYFERENCE in static and dynamic networks and how does it compare to the existing network inference algorithms? (2) How does predictive performance of DYFERENCE change with the number of cascades? (3) How does running time of DYFERENCE compare to the baselines? And, (4) How does DYFERENCE perform for the task of predicting diffusion and influential nodes?

**Baselines.** We compare the performance of DYFERENCE to NETINF [19], NETRATE [21], TOPOL-STM [13], DEEPCAS [15], EMBEDDEDIC [13] and INFOPATH [22]. INFOPATH is the only method able to infer dynamic networks, hence we can only compare the performance of DYFERENCE on dynamic networks with INFOPATH.

**Evaluation Metrics.** For performance comparison, we use Precision, Recall, F1 score, Map@$k$ and Hit@$k$. Precision is the fraction of edges in the inferred network present in the true network, Recall is the fraction of edges of the true network present in the inferred network, and F1 score is $2\times$(precision$\times$recall)/(precision+recall). MAP@$k$ is the classical mean average precision measure and Hits@$k$ is the rate of the top-$k$ ranked nodes containing the next infected node.

In all the experiments we use a sample size of $q = |E_c| - 1$ for all the cascades $c \in C$. We further consider a window of length $w = 1$ day in our dynamic network inference experiments in Fig 1 and $w = 2$-years in Table 3.

**Synthetic Experiments.** We generated synthetic networks consist of 1024 nodes and about 2500 edges using Kronecker graph model [35]: core-periphery network (CP) (parameters [0.9,0.5;0.5,0.3]), hierarchical community network (HC) (parameters [0.9,0.1;0.1,0.9]), and the Forest Fire model [36]: with forward and backward burning probability 0.2 and 0.17. For dynamic networks, we assign a pattern to each edge uniformly at random from a set of five edge evolution patterns: Slab, and Hump (to model outlinks of nodes that temporarily become popular), Square, and Chainsaw (to model inlinks of nodes that update periodically), and Constant (to model long term interactions) [22]. Transmission rates are generated for each edge according to its evolution pattern for 100 time steps. We then generate 500 cascades per time step (1 day) on the network with a random initiator [10].

Figures 1a, and 1b compare precision, recall and F1 score of DYFERENCE to INFOPATH for online dynamic network inference on CP-Kronecker network with exponential edge transmission model, and HC-Kronecker network with Rayleigh edge transmission model. It can be seen that DYFERENCE outperforms INFOPATH in terms of F1 score as well as precision and recall on different network topologies in different transmission models. Figures 1c, 1d, 1e compare F1 score of DYFERENCE compared to INFOPATH and NETRATE for static network inference for varying number of cascades over CP-Kronecker network with Rayleigh and Exponential edge transmission model, and Forest

Table 1: Performance of DYFERENCE for diffusion prediction compared to DEEPCAS, TOPOLSTM, and EMBEDDEDIC on Twitter and Memes datasets (TOPOLSTM requires the underlying network).

| | | Twitter | | | Memes | |
|---|---|---|---|---|---|---|
| *MAP@k* | @10 | @50 | @100 | @10 | @50 | @100 |
| DEEPCAS | 9.3 | 9.8 | 9.8 | 18.2 | 19.4 | 19.6 |
| TOPOLSTM | 20.5 | 20.8 | 20.8 | 29.0 | 29.9 | 30.0 |
| EMBEDDED-IC | 12.0 | 12.4 | 12.5 | 18.3 | 19.3 | 19.4 |
| DYFERENCE | **20.6** | **20.8** | **20.9** | **29.4** | **31.5** | **32.4** |
| *Hits@k* | @10 | @50 | @100 | @10 | @50 | @100 |
| DEEPCAS | 25.7 | 31.1 | 33.2 | 43.9 | 60.5 | 70.0 |
| TOPOLSTM | 28.3 | 33.1 | 34.9 | **50.8** | 69.5 | 76.8 |
| EMBEDDEDIC | 25.1 | 33.5 | 36.6 | 35.1 | 56.0 | 65.0 |
| DYFERENCE | **30.0** | **34.3** | **36.7** | 47.4 | **71.0** | **84.0** |

Table 2: Top 10 predicted influential websites of Memes (Linkedin) on 30-06-2011. The correct predictions are indicated in bold.

| DYFERENCE | INFOPATH |
|---|---|
| **pressrelated.com** | podrobnosti.ua |
| arsipberita.com | scribd.com |
| **news.yahoo.com** | derstandard.at |
| **in.news.yahoo.com** | heraldonline.com |
| podrobnosti.ua | startribune.com |
| **article.wn.com** | canadaeast.com |
| **ctv.ca** | news.yahoo.com |
| **fair-news.de** | proceso.com.mx |
| fanfiction.net | **article.wn.com** |
| bbc.co.uk | prnewswire.com |

Table 3: Performance of DYFERENCE for dynamic bankruptcy prediction compared to INFOPATH on financial transaction network from 2010 to 2016. In 2010, a financial crisis hit the network.

| | | 2012 | | | 2014 | | | 2016 | |
|---|---|---|---|---|---|---|---|---|---|
| *MAP@k* | @10 | @20 | @30 | @10 | @20 | @30 | @10 | @20 | @30 |
| INFOPATH | 4.0 | 5.3 | 6.6 | 35.0 | 34.5 | 30.0 | 54.7 | 65.0 | 65.0 |
| DYFERENCE | **17.6** | **19.1** | **20.6** | **62.0** | **51.9** | **38.1** | **69.6** | **85.7** | **85.7** |
| *Hits@k* | @10 | @20 | @30 | @10 | @20 | @30 | @10 | @20 | @30 |
| INFOPATH | 20.0 | 25.0 | 26.6 | 50.0 | 55.0 | 50.0 | 80.0 | 65.0 | 65.0 |
| DYFERENCE | **40.0** | **45.0** | **46.6** | **70.0** | **65.0** | 50.0 | **80.0** | **70.0** | **70.0** |

Fire network with Power-law edge transmission model. We observe that DYFERENCE consistently outperforms the baselines in terms of accuracy and is robust to varying number of cascades.

**Real-world Experiments.** We applied DYFERENCE to three real wold datasets, (1) Twitter [37] contains the diffusion of URLs on Twitter during 2010 and the follower graph of users. The network consists of 6,126 nodes and 12,045 edges with 5106 cascades of length of 17 on average, (2) Memes [38] contains the diffusion of memes from March 2011 to February 2012 over online news websites; The real diffusion network is constructed by the temporal dynamics of hyperlinks created between news sites. The network consists of 5,000 nodes and 313,669 edges with 54,847 cascades of length of 14 on average, and (3) a European county's financial transaction network. The data is collected from the entire country's transaction log for all transactions larger than 50K Euros over 10 years from 2007 to 2017, and includes 1,197,116 transactions between 103,497 companies. 2,765 companies are labeled as bankrupted with corresponding timestamps. In 2010, a financial crisis hit the network. For every 2 years from 2010, we built a diffusion of bankruptcy with average length of 85 between 200 bankrupted nodes that had the highest amount of transactions each year.

Figures 1g, 1h, 1i, 1j compare the F1 score of DYFERENCE to INFOPATH for online dynamic network inference on the time-varying hyperlink network with four different topics over time from March 2011 to July 2011. As we observe, DYFERENCE outperforms INFOPATH in terms of the prediction accuracy in all the networks. Figure 1f compares the running time of DYFERENCE to that of INFOPATH. We can see that DYFERENCE has a running time that is comparable to INFOPATH, while consistently outperforms it in terms of the prediction accuracy.

**Diffusion Prediction.** Table 1 compares Map@$k$ and Hits@$k$ for DYFERENCE vs. TOPOLSTM, DEEPCAS, and EMBEDDEDIC. We use the infection times in the first 80% of the total time interval for training, and the remaining 20% for the test. It can be seen that DYFERENCE has a very good performance for the task of diffusion prediction. Note that TOPOLSTM needs complete information about the underlying network structure for predicting transmission probabilities, and INFOPATH relies on predefined parametric probability distributions for transmission rates. On the other hand, DYFERENCE does not need any information about network structure or transmission rates.

Table 3 compares Map@$k$ and Hits@$k$ for DYFERENCE vs. INFOPATH for dynamic bankruptcy prediction on the financial transaction network since the crisis hit the network at 2010. We used windows of length $w = 2$ years to build cascades between bankrupted nodes and predict the

companies among the neighbors of the bankrupted nodes that are going to get bankrupted the next year. It can be seen that DYFERENCE significantly outperforms INFOPATH for bankruptcy prediction.

**Influence Prediction.** Table 2 shows the set of influential websites found based on the predicted dynamic Memes network by DYFERENCE vs INFOPATH. The dynamic Memes network for Linkedin is predicted till 30-06-2011, and the influential websites are found using the method of [39]. We observe that using the predicted network by DYFERENCE we could predict the influential nodes with a good accuracy.

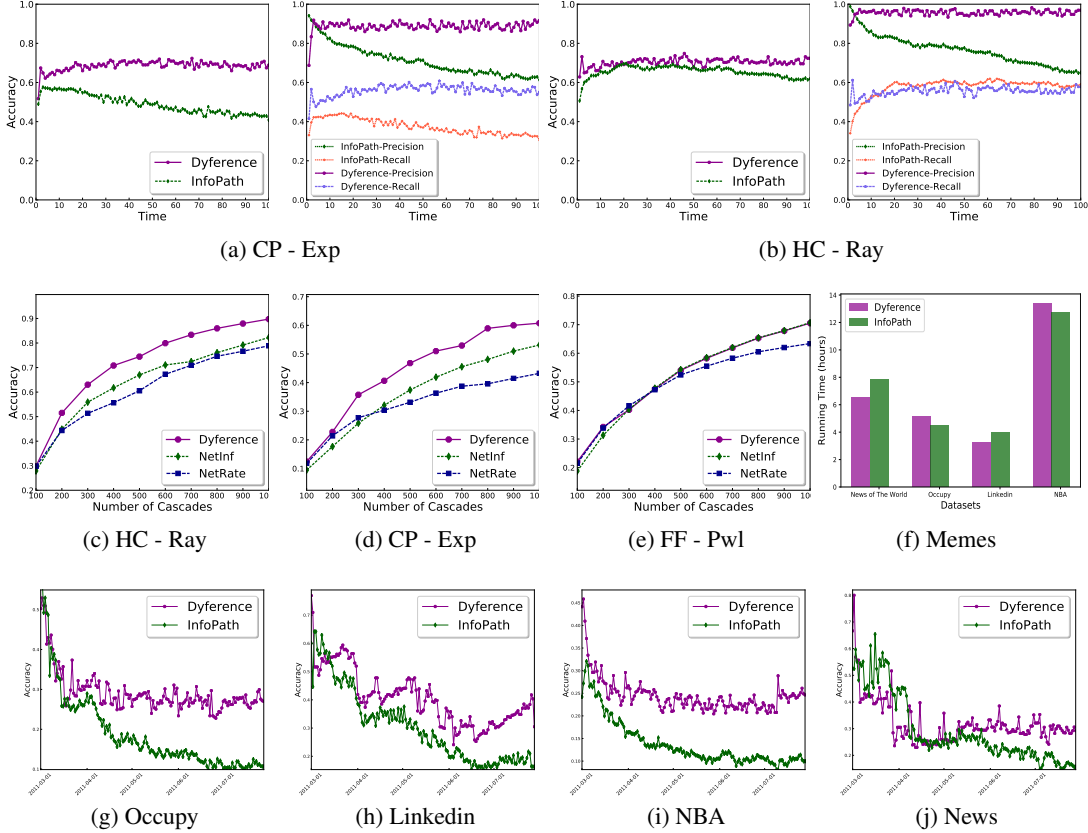

(a) CP - Exp            (b) HC - Ray

(c) HC - Ray      (d) CP - Exp      (e) FF - Pwl      (f) Memes

(g) Occupy      (h) Linkedin      (i) NBA      (j) News

Figure 1: Precision, Recall and F1 score of DYFERENCE. **(a)** Compared to INFOPATH for dynamic network inference over time on Core-Periphery (CP) Kronecker network with exponential transmission model, and **(b)** Hierarchical (HC) Kronecker network with Rayleigh transmission model. **(c)** accuracy of DYFERENCE compared to INFOPATH and NETRATE for static network inference for varying number of cascades over CP-Kronecker network with Rayleigh, and **(d)** Exponential transmission model, and **(e)** on Forest Fire network with Power-law transmission model. **(f)** compares the running time of DYFERENCE with INFOPATH for online dynamic network inference on the time-varying hyperlink network with four different topics Occupy with 1,875 sites and 655,183 memes, Linkedin with 1,035 sites and 155,755 memes, NBA with 1,875 sites and 655,183 memes, and News with 1,035 sites and 101,836 memes. **(g), (h), (i), (j)** compare the accuracy of DYFERENCE to INFOPATH for online dynamic network inference on the same dataset and four topics from March 2011 to July 2011.

## 6 Conclusion

We considered the problem of developing generative dynamic network models from partial observations, i.e. diffusion data. We proposed a novel framework, DYFERENCE, for providing a non-parametric edge-exchangeable network model based on a mixture of coupled hierarchical Dirichlet processes (MDND). However, our proposed framework is not restricted to MDND and can be used along with any generative network models to capture the underlying dynamic network structure from partial observations. DYFERENCE provides online time-varying estimates of probabilities for all the potential edges in the underlying network, and track the evolution of the underlying community structure over time. We showed the effectiveness of our approach using extensive experiments on synthetic as well as real-world networks.

**Acknowledgment.** This research was partially supported by SNSF P2EZP2_172187.

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
