[Reviews · NeurIPS 2018]

Reviewer 1



The paper proposes to use a generative model to infer dynamic networks from partial observations (cascades of information, such as twitter memes). The algorithm is derived and tested on both simulation and real data. Overall it is an interesting approach and merits publication, though it ultimately performs quite similarly to TopoLSTM in Table 1 and InfoPath on Figure 1f. I think it'd also be nice to find a more contemporary killer application, rather than working with an almost decade old data, but it's a minor concern. Quality: The work builds on (Williamson, 2016) and seems solid. The authors don't cite some of the dynamic and exchangeable models mentioned in a survey (A Goldenberg, A Zheng, S Fienberg and E Airoldi (2010) A Survey of Statistical Network Models, Foundations and Trends in Machine Learning, Vol 2 (2): pp 129-233) that seem relevant to their work. Again this is a minor concern. I have one concern about the reliance on the DP models to generate edges from the observations. It was previously shown that if node degree follows power law lead to edges following piecewise power law distribution (Wang X, Chen Z, Liu P, Gu Y. Edge balance ratio: Power law from vertices to edges in directed complex network. IEEE Journal of Selected Topics in Signal Processing. 2013 Apr;7(2):184-94.). As such I'm not sure how generalizable this model is for real networks. The simulation models used also have similar properties, hence the proposed method performs fairly well. On the real world data, the performance is similar to some of the other approaches as I mentioned above. It is still a work of solid quality, just somewhat concerned about generalizability of the underlying assumptions. Clarity: The paper is very clear. It is almost unnecessary to include the provided algorithms, especially Algorithm 3, but for completeness, these descriptions fit quite well. Originality: though the work follows on some of the existing literature, I believe that it is substantially original and novel to score highly on this metric. Significance: I'm not entirely convinced about the significance. The focus is of course on the method, which is substantially novel, but the experiments were ran on some of the older data, failing to excite the reader with a killer application.

Reviewer 2



The authors consider the problem of constructing generative models from partial observations. More specifically, they propose a generative model for networks where the observations consist of times when cascades spread over the network. In this setting, the observations lack information regarding the resulted edges. The goal is to infer the latent structure of the network over which the cascades propagated. They propose a hierarchical non-parametric edge-exchangeable network model along with a inference framework, DYFERENCE. I really enjoyed reading this work. The authors deal with a very interesting problem with possible real world applications. The proposed work is novel enough to my understanding. They do an excellent job presenting their idea in a coherent and scientifically deep fashion. The inference seems the most challenging part of the model but the proposed framework covers the difficult aspects of it. More: The content appear to be correct regarding technical details. The submission is very clear and well organised. The experimental results are thorough and sufficiently prove the performance of the model. Line 119: not sure how the integer valued weights are sampled here. Also, z_ij and d_uv in eq(3), why different term here? All in all, I am eager to suggest acceptance.

Reviewer 3



The authors propose a framework for modeling dynamic network cascade diffusion and contrast the approach in particular to InfoPath but also several more recent static network estimation procedures. The model is probabilistic and extends the work of Williams (ref 7) to cascade diffusion by learning the latent network structure solely based on the nodes observed contamination times within a cascade. The framework uses the partial observational framework discussed in ref 27 and inference of determinantal point processes as discussed in ref 28 in order to establish a distribution of plausible edges given the observed cascades. The MDMN model of ref 7 is then used as a latent representation of this unobserved network representation. The framework is extended to the modeling of dynamic networks considering a time interval w and updating the model at time t+1 using the model at time t as a starting point. Quality: The manuscript is well written, the experimentation solid, and the approach well motivated improving upon current state-of-the-art of network quantification based on observed cascades. Clarity: The manuscript is generally well written and well rooted in the existing literature. There are a few typos that should be corrected, i.e. “is the |Rk x Rk| is the restriction…” -> “is the |Rk x Rk| restriction…” “and and” -> “and” I would further recommend that the section “Related Work” be integrated into the existing introduction as there are quite some overlaps between the introduction and related work section. In particular, remarks such as “Our work is different in nature to the existing methods in that we aim at providing a generative probabilistic model for the underlying dynamic network from diffusion data.” and “However, these models assume that networks are fully observed. In contrast, our work here considers the network is unobserved but what we observe are node infection times of a stochastic cascading process spreading over the network.” These points are already remarked in the introduction also discussing some of the current state of the art. It could on related work also be worth mentioning probabilistic generative dynamic models assuming fully observed networks, see for instance as part of the related work Ishiguro, Katsuhiko, et al. "Dynamic infinite relational model for time-varying relational data analysis." Advances in Neural Information Processing Systems. 2010. Originality: The novelty of the paper is combining the use of latent network modeling based on the mixture of Dirichlet network distributions (MDND) modeling framework ref 7 to operate not directly on an observed network but a latent network defined though cascades using results from ref. 27 and 28 in order to define a distribution over latent edges. In ref 7 the MDND was established as a competitive link-prediction framework and it is thus reasonable to use this as a starting point of latent variable modeling of the network structure. However, other competitive probabilistic link prediction modeling approaches choices could also have been relevant choices, such as: Palla, Konstantina, David Knowles, and Zoubin Ghahramani. "An infinite latent attribute model for network data." arXiv preprint arXiv:1206.6416 (2012). And within the framework of Fox and Carron the extension to modeling block structure Herlau, Tue, Mikkel N. Schmidt, and Morten Mørup. "Completely random measures for modelling block-structured sparse networks." Advances in Neural Information Processing Systems. 2016. It would improve the paper to discuss how the present framework extends to arbitrary generative models replacing the MDND with other choices of latent representations which would expand the scope of the present framework. That being said there are many merits of the MDND and this is a sound and strong starting point. Combining generative network models for modeling cascades is an interesting and useful novel contribution. Significance: The proposed framework is a fine extension of the MDND framework to the modeling of network structure based on observed cascades and form a new and powerful tool for analyzing network dynamics. In particular, the experimental results emphasizes the utility of the proposed framework outperforming existing approaches while providing a nice probabilistic generative modeling framework for the extraction of dynamic network structure. It is unclear to me why the network needs to be discretized in time segments w as the cascades are in continuous time. Furthermore, the dynamic modeling stemming from updating the model from previous timesteps seem a bit heuristic and to hinge on how much the sampling procedure changes the model across the time steps which in turn also reflect the ability of the sampler to properly mix and for how long the inference procedure is run (i.e., how many samples are drawn at each time step). It would improve the manuscript to clarify these aspects. It would also be good to include a discussion of the stability of the inferred parameters and thus reliability of the inference procedure. Furthermore, it would be good to clarify how the predictions are made, i.e. InfoPath rely on point estimates but the current procedure I assume is based on averaging samples. This should be clarified, and if averaging samples it would be good to also demonstrate what the performance of a corresponding point estimate (by highest likelihood sample) would provide to assess the merits of simply providing Bayesian averaging as compared to a more advanced MDND model. I have read the authors rebuttal and find that they satisfactorily discuss the raised concerns. I argue for accepting this paper.